# Prevalence of Advanced Parkinson’s Disease in Patients Treated in the Hospitals of the Spanish National Healthcare System: The PARADISE Study

**DOI:** 10.3390/brainsci11121557

**Published:** 2021-11-24

**Authors:** Juan Carlos Martínez-Castrillo, Pablo Martínez-Martín, Ángel Burgos, Gloria Arroyo, Natalia García, María Rosario Luquín, José Matías Arbelo

**Affiliations:** 1Departamento de Neurología, Hospital Universitario Ramón y Cajal, Ctra. de Colmenar Viejo, km. 9100, 28034 Madrid, Spain; 2Centro de Investigación Biomédica en Red Sobre Enfermedades Neurodegenerativas (CIBERNED), Instituto de Salud Carlos III (ISCIII), Calle Valderrebollo, 5, 28031 Madrid, Spain; pmm650@hotmail.com; 3Pivotal, S.L.U. Calle Gobelas, 19, 28023 Madrid, Spain; Angel.Burgos@pivotalcr.com; 4AbbVie Spain S.L.U. Avenida De Burgos 91, 28050 Madrid, Spain; gloria.arroyo@abbvie.com (G.A.); natalia.garcia@abbvie.com (N.G.); 5Servicio de Neurología, IdiSNA, Clínica Universidad de Navarra (CUN), Av. de Pío XII, 36, 31008 Pamplona, Spain; rluquin@unav.es; 6Servicio de Neurología, Hospital Universitario San Roque Las Palmas, Calle Dolores de la Rocha 5, 35001 Las Palmas de Gran Canaria, Spain; jmarbelo@gmail.com

**Keywords:** incidence, Advanced Parkinson’s disease, prevalence, idiopathic Parkinson’s disease, quality of life

## Abstract

Background: Advanced Parkinson’s disease (APD) has been recently defined as a stage in which certain symptoms and complications are present, with a detrimental influence on the overall patient’s health conditions and with a poor response to conventional treatments. However, historically, the term APD has been controversial, thus consequently, APD prevalence has not been previously studied. Objectives: The main objective was to determine the prevalence of APD in patients diagnosed with idiopathic PD in hospitals of the Spanish National Healthcare System. Secondary objectives were the prevalence and incidence of PD and the clinical and sociodemographic characteristics and quality of life of patients with APD or non-APD. Methods: This was a non-interventional, cross-sectional, multicenter, national study in the hospital setting. Results: The study population included 929 patients with PD (mean age 71.8 ± 10.1 years; 53.8% male) and a mean time since diagnosis of 6.6 ± 5.4 years. At the time of diagnosis, 613 patients (66.06%) reported having had premotor symptoms. The Hoehn and Yahr stage was 1 in 15.7% of the patients, 2 in 42.8%, 3 in 30.1%, 4 in 9.9%, and 5 in 1.4%; 46.9% of the patients had comorbidities (mean age-adjusted Charlson comorbidity index 3.5 ± 1.7; median 10-year survival 77%) and the mean 8-item Parkinson’s Disease Quality of Life Questionnaire was 27.8 ± 20.5. We found an APD prevalence of 38.21% (95%CI: 35.08–41.42%), a PD prevalence of 118.4 (95%CI: 117.3–119.6), and a PD incidence of 9.4 (95%CI: 5.42–13.4) all per 100,000 population. Among the APD population, a 15.2% were receiving some form of therapy for advanced stages of the disease (deep brain stimulation, levodopa/carbidopa intestinal gel, or apomorphine subcutaneous infusion). Conclusions: The percentage of patients with APD in the hospitals of the Spanish National Healthcare System was 38.2%.

## 1. Introduction

Parkinson’s disease (PD) is one of the most prevalent neurodegenerative disorders in the world [1] and a growing source of disability globally [2]. Prevalence rates for patients with PD vary across studies and countries; however, a meta-analysis published in 2014 estimated a prevalence rate of PD worldwide of 300 patients per 100,000 population [3].

Parkinson’s disease is a progressive disease; most patients progress to an advanced stage after 7–10 years form diagnosis [4]. In this advanced stage, the disease is highly disabling, has a complex management, impacts seriously patient’s quality of life (QoL), and increases healthcare system expenses. Typically, advanced Parkinson’s disease (APD) patients develop motor fluctuations, dyskinesias, and non-motor symptoms within 3 to 5 years of starting dopaminergic therapy, and these motor complications could be, at this stage, refractory to treatment. Conversely, comorbidities within this population, arising independently of the underlying disease, such as neoplasms [5], may increase the burden of the social and Healthcare System. However, despite this clinical and administrative relevance, the prevalence of advanced PD is unknown, mainly because of the absence of an APD-specific definition. Sizing the number of patients at this advance stage of the disease should help the health authorities to reorganize resources, if necessary.

In this epidemiologic study, the main objective was to determine the prevalence of APD in patients diagnosed with idiopathic PD in hospitals of the Spanish National Healthcare System. Secondary objectives included the assessment of the prevalence and incidence of PD in hospitals of the Spanish National Healthcare System, the description of the clinical and sociodemographic characteristics of patients with PD, and the comparison of the QoL of patients with APD and non-APD (nAPD).

## 2. Materials & Methods

### 2.1. Study Design

This was a non-interventional, cross-sectional, multicenter, national study carried out in Spain from April 2017 through January 2019 at 21 participant clinical sites. The study population consisted of patients with idiopathic PD according to the UK Parkinson’s Disease Society Brain Bank criteria [6] at the participant sites. These sites were randomly selected from the list of the 234 Spanish public hospitals and stratified according to the type of hospital, considering the total population covered by the site. In Spain, based on the time and resources dedicated to PD, three types of neurology services at the hospital level can be described (in descending order): (1) services with a movement disorders unit; (2) services with PD dedicated clinics; (3) services with general neurology clinics. PD dedicated clinics are general neurology services with at least one neurologist attending a specific clinic for PD with some regularity. Out of a total population of 46,423,064 inhabitants in Spain at the time of the study design [7], the Spanish public hospitals covered 42,501,454 Spanish people: 8,798,286 (20.7%) at hospitals with movement disorder units; 17,076,616 (40.2%) at hospitals with PD dedicated clinics; 16,626,552 (39.1%) at hospitals with general neurology services [8]. Subsequently, to guarantee the representativity of the sample, approximately 20% of the selected patients should be treated at hospitals with movement disorder units (MDUs), 40% at hospitals with PD dedicated clinics, and 40% at hospitals with general neurology services. When a center denied its participation, it was randomly substituted by another with the same characteristics. At each center, the selection of the patients was performed using systematic random sampling. The stratified random sampling of sites, the determination of patient sample size, and the estimation (±3) of the prevalence followed the methods as described in Levy and Lemeshow [9]. The random sampling (of both centers and patients) guaranteed that patients with PD entering the study were representative of the whole Spanish PD population. The APD diagnosis was made according to the neurologist’s criteria, which was considered the gold standard, and with the Questionnaire for Advanced Parkinson’s Disease (CDEPA), a tool that was developed to assist in the diagnosis of APD [10]. This tool was developed following the Delphi method; APD was defined as a stage of PD in which certain symptoms and complications are present, with a detrimental influence on the overall patient’s health conditions and poor response to conventional treatment [11]. In case of a discrepancy, the neurologist’s criteria prevailed. Patients provided signed informed consent to participate in the study. Patients unable to provide signed informed consent and whose caregiver was unwilling to provide written informed consent on their behalf were excluded.

All patients with PD from the selected hospitals were registered for determining the prevalence and the incidence of PD. Patients with APD were identified among those randomly selected. For patients who chose not to participate in the study, the reason was recorded, and other patients were recruited instead. Because the frequency of visits differs across hospitals (from 3 times per year to once per year), the study recruitment period was established in 1 year to ensure that all patients being cared for in a particular center were registered. In few centers in which the recruitment period was shorter than 12 months, PD prevalence and incidence were annualized and weighted according to the number of months of recruitment to make them comparable.

The study was conducted in accordance with ethical principles that have their origin in the Declaration of Helsinki. The study was approved by the Spanish Agency of Medicines and Medical Devices and the Ethics Committees of all the participant hospitals.

### 2.2. Variables

Sociodemographic data and clinical variables (age, sex, occupational status, PD disease duration, Hoehn and Yahr (H&Y) stage at the time of PD diagnosis and at the time of patient inclusion in the study, pre-motor symptoms, and concurrent comorbidities) were obtained from computerized hospital records, patient medical records, or discharge reports. APD diagnosis was established according to the criteria above mentioned. The percentage of patients with APD was calculated by dividing the total number of patients with APD per the total number of patients with PD recruited in the study and multiplying by 100. The following equation was used to estimate PD annual prevalence: 

(Total number of patients with PD in a 12-month period/total number of inhabitants covered by the sites) × 100,000.

The following equation was used to estimate PD annual incidence: 

(Total number of new cases of PD in a 12-month period/total number of inhabitants covered by the sites) × 100,000.

Ten-year survival was estimated using the Charlson comorbidity index [12]. Patient QoL was evaluated using the 8-item Parkinson’s Disease Quality of Life Questionnaire (PDQ-8; the score was standardized on a scale of 0 to 100) [13]. Additionally, the H&Y stage in ON or OFF state was assessed using the H&Y rating scale [14].

### 2.3. Statistical Analysis

Sample size calculation is presented as Supplementary Material.

Descriptive statistics (mean, standard deviation, median, percentage, and 95% confidence interval) were used as needed. For comparisons, and considering their assumptions for use, Student’s *t* or Wilcoxon–Mann–Whitney tests were applied.

Chi-square test or Fisher exact test were used, as applicable to assess possible association between qualitative variables. McNemar’s test or Cochran’s Q test were used for paired data. Unweighted kappa statistic was used to assess the concordance between the neurologist’s criteria and the CDEPA identifying APD patients. Logistic regression was performed to ascertain the effects of the presence of CDEPA APD definitory symptoms, age, sex, age-adjusted Charlson comorbidity index, and H&Y stage on the likelihood that APD patients receive a device aided therapy (DAT), that is, deep brain stimulation, levodopa/carbidopa intestinal gel, or apomorphine subcutaneous infusion. A two-tailed *p* < 0.05 was used to denote statistically significant differences. All analyses were performed using SAS^®^ v9.4 software (SAS Institute Inc., Cary, NC, USA).

## 3. Results

Nine hundred and fifty-five patients were invited to participate in the study, of whom 26 (2.8%) did not provide consent for study participation. Thus, the study population included 929 patients: 227 (24.4%) in hospitals with a movement disorder unit, 297 (32.0%) in hospitals with a PD dedicated clinic and 405 (43.6%) in hospitals with a general neurology service. Clinical and sociodemographic characteristics of patients at the time of study inclusion according hospital type are shown in Table 1. 

Among the 929 patients, 355 were identified as having APD. Thus, the prevalence of APD in patients with PD attended at hospitals of the Spanish National Health System was 38.2% (95%CI: 35.1–41.4%). No statistically significant differences were seen in the percentages of patients with APD among those patients with PD attended at hospitals with general neurology services (38.1% (95%CI: 33.3–42.9%)), with PD dedicated clinics (37.7% (95%CI: 32.2–43.5%)), and with movement disorder units (39.2% (95%CI: 32.8–45.9%)) (*p* = 0.94).

The concordance between the neurologist’s criteria and the CDEPA identifying patients with APD was κ = 0.94 (95%CI: 0.92–0.96; *p* < 0.0001). A high level of concordance was also found between the neurologist’s criteria and the CDEPA when concordance was assessed according to the type of hospital, as follows: hospitals with general neurology services κ = 0.90 (95%CI: 0.86–0.94; *p* < 0.0001), with PD dedicated clinics κ = 0.97 (95%CI: 0.94–1.00; *p* < 0.0001), and with movement disorder units κ = 0.97 (95%CI: 0.94–1.00; *p* < 0.0001). 

Differences were found when comparing the sociodemographic and clinical characteristics of patients with nAPD and APD (Table 2). At the time of inclusion in the study, mean age was higher in patients with APD (73.7 ± 10.0 years) than in patients with nAPD (70.7 ± 10.0 years; *p* < 0.0001), a higher percentage of men (56.8%) was found in patients with nAPD than in those with APD (49.0%; *p* = 0.0208), and a higher H&Y stage evaluated in patients considered ON at the time of diagnosis was found in patients with APD vs. nAPD (*p* < 0.0001). Additionally, a higher percentage of patients with APD (83.0%) than nAPD (72.3%) reported having had premotor symptoms (*p* = 0.0007), mean age-adjusted Charlson comorbidity index was significantly higher in patients with APD (3.77 ± 1.78) than in those with nAPD (3.27 ± 1.58; *p* < 0.0001), and the median 10-year survival probability was significantly lower in patients with APD (53%) than in those with nAPD (77%; *p* < 0.0001).

Reasons for not having started a DAT among patients with APD also differed according to hospital type (*p* = 0.028; Figure 1). Patients with a disease duration of ≥10 years were 6.77 times (95%CI: 2.83–16.17) more likely to receive a DAT than patients with a disease duration of <10 years; patients with motor fluctuations with an OFF time of >25%, with limitation to perform basic activities but without requiring help, were 2.11 times (95%CI: 1.01–4.41) more likely to receive a DAT than patients who did not have motor fluctuations with an OFF time of >25%, with limitation to perform basic activities but without requiring help. Increasing age was also associated with an increased likelihood of receiving a DAT (odds ratio 1.10; 95%CI: 1.06–1.14).

The QoL of patients with PD, assessed with the PDQ-8 at the time of study inclusion, showed a mean score of 27.8 ± 20.5. No statistically significant differences were found when mean PDQ-8 scores obtained in patients receiving care from different hospital types were compared. However, a statistically significant worsening in QoL was found in patients with APD (Figure 2). The mean PDQ-8 score in patients with nAPD was 19.2 ± 15.3 versus 41.7 ± 20.3 in APD patients (*p* < 0.0001). This difference was also observed in all subdomains of the questionnaire. Among the APD population, there were no significant differences in PDQ-8 scores between patients treated with a DAT compared with those not treated with a DAT (43.0 ± 22.4 vs. 41.5 ± 19.9, respectively; *p* = 0.62). However, there were differences in PDQ-8 scores between the different DAT untreated patient populations based on the reasons for not having started a DAT (in order of increasing score): 34.9 ± 17.5 (clinically stable), <39.2 ± 19.3 (option not yet considered), <42.7 ± 21.1 (on waiting list), <43.1 ± 16.2 (treatment rejected by the patient), <49.5 ± 12.3 (other reasons), and <57.0 ± 24.0 (DAT contraindicated), *p* < 0.0001.

The prevalence of patients diagnosed with PD at hospitals of the Spanish National Healthcare System was 118.4 patients (95%CI: 117.3–119.6) per 100,000 population. The annual incidence of patients with PD at hospitals of the Spanish National Healthcare System was 9.4 patients (95%CI: 5.42–13.4) per 100,000 population. 

## 4. Discussion

In this study, the percentage of patients with APD among 929 patients with PD cared for in the hospitals of the Spanish National Healthcare System was 38.2%. This percentage was slightly lower than the 51.3% found in the OBSERVE-PD study that was a cross-sectional, observational, multicenter, multicountry study carried out in 128 movement disorder centers from 18 countries involving 2615 patients [15]. In the OBSERVE-PD study, the percentages of patients with APD varied regionally and ranged from 24% to 82%; this broad range could reflect differences between countries or study populations. Our study was conducted in a single country, and patients and hospitals were randomly selected; thus, the sample of participant patients were representative of the Spanish PD patient population. The fact that the sites participating in the OBSERVE-PD study were movement disorder centers does not seem to explain the higher percentages of patients with APD found in that study, as we found no significant differences between the different hospital types. Clinical and sociodemographic differences between patients with APD and nAPD were found in both the OBSERVE-PD and PARADISE studies. To the best of our knowledge, no other studies have been published analyzing the prevalence of APD and the clinical and sociodemographic characteristics of these patients.

In our study, a higher proportion of men was found among patients with nAPD; however, patient age, disease duration, H&Y stage, proportion of patients with premotor symptoms at diagnosis, and age-adjusted Charlson comorbidity index were higher and 10-year survival probability was lower among patients with APD. In the OBSERVE-PD study, patients with APD had a higher disease duration, needed more caregiver support, presented with more motor fluctuations, and had a higher score in the Unified Parkinson’s Disease Rating Scale II, III, and V than patients with nAPD [15]. Notably, the percentage of patients receiving some of the therapies considered for APD was significantly lower in our study than in the OBSERVE-PD (15.2% vs. 44%, respectively). In order to improve patient’s care, this data deserves particular focus and reasons need to be analyzed. Problems such as availability of DAT, lack of prescription for some other reason or difficulties to identify DAT candidates by the neurologist could be some of them.

The QoL was significantly worse among patients with APD compared with those with nAPD in our study, with PDQ-8 scores of 41.7 and 19.2, respectively. These results were quite similar to those in the OBSERVE-PD study, where PDQ-8 scores were 36.6 and 20.7 in patients with APD and nAPD, respectively [15]. No significant differences in the QoL of patients cared for in different types of hospitals were found in our study. However, surprisingly, QoL was not different between APD DAT-treated patients and APD DAT-nontreated patients, although the three current DAT have demonstrated significant improvements in QoL of APD patients [16,17,18]. To explain these discrepancies, it is necessary to take into consideration that our study is a cross-sectional study and the average treatment duration of the DAT in this population was approximately 3 years.

The prevalence of PD can be influenced not only by the incidence of the disease, the survival and age of the study population, and genetic and environmental factors, but also by the study methodology and the criteria applied for disease diagnosis [19]. In our study, we found a PD prevalence of 118.4 per 100,000 population. This prevalence is much lower than the prevalence ranging from 901 to 1500 patients with PD per 100,000 population found in three studies conducted in small geographic areas of Spain, using door-to-door methodology, and in an elderly population [20,21,22]. However, the prevalence found in our PARADISE study is slightly lower than the prevalence ranging from 161.5 to 277 per 100,000 population found in six other studies carried out in Spain [23,24,25,26,27,28]. These studies were also conducted in small geographic areas, two of which were population-based studies in patients with PD requiring health care services [24,25], and the other four determined the prevalence of PD by use/consumption of antiparkinsonian drugs [23,26,27,28]. In 2005, von Campenhausen et al. carried out a systematic literature search to identify studies on the prevalence and incidence of PD in European countries. These authors found a PD prevalence in Europe ranging from 65.6 to 12,500 per 100,000 population [29]. In 2014, Pringsheim et al. found a worldwide PD prevalence of 315 patients (95% CI: 113–873) per 100,000 population in a meta-analysis that included all published studies of door-to-door surveys or random population samples with a physical examination by a health professional to confirm or exclude a diagnosis of PD [3].

Concerning the incidence of PD, we found a PD annual incidence of 9.4 per 100,000 population. In 2003, Twelves et al. carried out a systematic review and found 25 studies focused on the incidence of PD. The incidence ranged from 1.5 per 100,000 per year in China to 26 per 100,000 per year in the United Kingdom. All studies included in the review were methodologically different, but five were similar enough to perform a comparison. Four of them found a standardized incidence of 16 to 19 per 100,000 per year, but the fifth study, carried out in Italy, found an incidence of 8.4 per 100,000 population [30]. In the above-mentioned door-to-door study conducted in central Spain [20], in a population aged 65 to 85 and over years and adjusted to the standard European population, the average annual incidence rate for PD was 186.8 per 100,000 person-years. Notably, the 53.3% of the patients were detected through the screening and had not been diagnosed previously. This could explain why the prevalence and incidence rates from hospital-based studies are lower. In our hospital-based study, there was no age limit for recruitment. As previously mentioned, differences with our study could be explained by environmental or genetic factors, geographic distribution, differences in populations, or different methodological approaches, including diagnostic criteria.

This study has some limitations. The real prevalence of PD in Spain could be higher than the prevalence found in our study because there can be patients who have PD but have not yet been diagnosed. The only way to eliminate this bias would be to conduct population studies “door to door”, in which specialists undertake the diagnosis in a selected sample of subjects. These types of studies are extremely expensive; thus, it is practically unviable to carry them out at national level. Moreover, as PD diagnosis is mainly clinical and neuroimaging testing were not required in the study to recruit patients, it is possible that patients currently diagnosed as having PD actually have another disorder under the category of parkinsonism, quite likely progressive supranuclear paralysis with predominant parkinsonism [31]. Although misdiagnosed patients are few, ranging from 5.9% [32] to 6.6% [33], this could have happened in our study.

Inaccuracy in PD diagnosis may also affect the assessment of the incidence of the disease. It has been published that specialists can make an erroneous initial diagnose of PD in 6% to 25% of the cases [32]. However, we should also consider that, although some patients can be diagnosed with PD, they may have an alternative parkinsonian condition; the reverse can also occur. In the study of Schrag et al., performed in the primary care setting, ≥15% of patients initially diagnosed with PD finally did not meet PD criteria; by contrast, 19% of patients who should have been diagnosed with PD were not [34].

In addition, our prevalence and incidence data are limited to the Spanish public hospitals, which means the Spanish population being cared for by private centers has not been included in our study. Nevertheless, we consider that this patient population exclusion had little impact on the results obtained as only 3.8% of the Spanish population were cared in private centers when the study begun [35]. Finally, it is also important to consider that the institutionalized population and potential PD population not yet diagnosed or diagnosed but still at the primary care level or in transit to the neurologist’s hospital have neither been included.

The strength of this study was the methodology used; the stratified random sampling that led to a selection of patients that represents the whole national territory; the large number of participating patients made it possible to obtain precise data with narrow confidence intervals; and the use of the validated objective tool CDEPA, which helped to identify APD patients.

## 5. Conclusions

In our study, the percentage of patients found with APD among those with PD cared for in hospitals of the Spanish National Healthcare System, and which is based on the neurologist’s criteria, is 38.2%. To the best of our knowledge, this is the first epidemiological study quantifying APD at the hospital level, including all types of neurology services in terms of specialization in PD. This prevalence is in concordance with that found by identifying patients with the screening tool CDEPA. Besides, the prevalence and the annual incidence of diagnosed PD found in this study is 118.4 and 9.4 per 100,000 population, respectively, which is lower than previously published. Conversely, we found a low percentage of APD patients receiving DAT, which would need further investigation.

## Figures and Tables

**Figure 1 brainsci-11-01557-f001:**
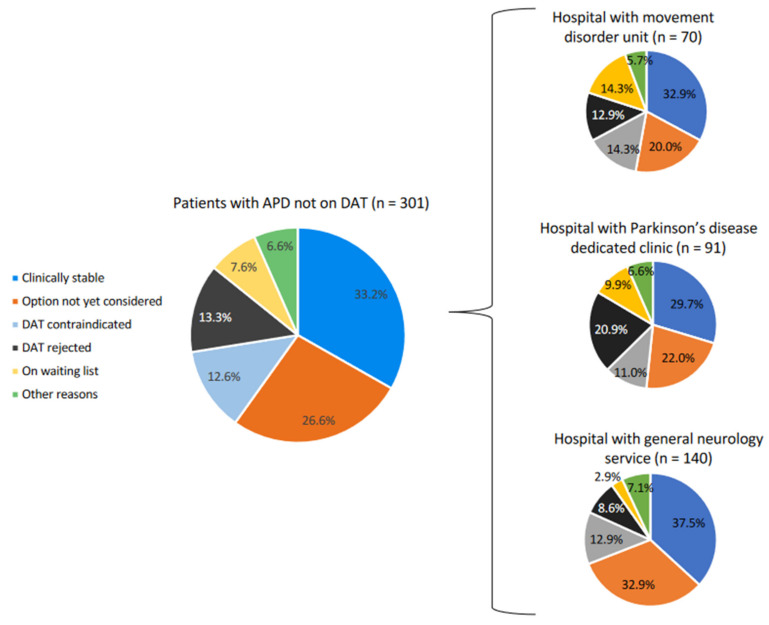
Reasons why patients with ADP were not on a DAT. ADP, advanced Parkinson’s disease; DAT, device-aided therapy. Statistically significant differences among hospital types (*p* = 0.028).

**Figure 2 brainsci-11-01557-f002:**
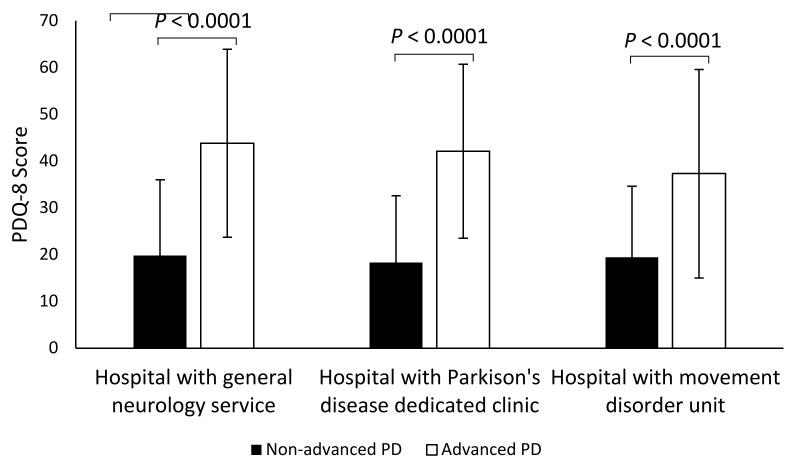
Quality of life of patients with advanced PD and non-advanced PD according to hospital type. PD, Parkinson’s disease; PDQ-8, 8-item Parkinson’s disease quality of life questionnaire. Bars represent mean ± SD.

**Table 1 brainsci-11-01557-t001:** Clinical and sociodemographic characteristics of patients at the time of study inclusion according hospital type.

Characteristic	Hospital with General Neurology Service (n = 405)	Hospital with Parkinson’s Disease Dedicated Clinic (n = 297)	Hospital with Movement Disorder Unit (n = 227)	Total (n = 929)	*p* Value
Age, years					<0.0001
Mean ± SD	73.5 ± 10.0	69.4 ± 9.8	72.0 ± 10.1	71.8 ± 10.1	
Median	75	70	72	73	
Q1–Q3	68–81	63–77	65–80	65–80	
Min–Max	36–92	38–90	42–93	36–93	
Sex, male, % ^a^					
30–49	72.7	83.3	60.0	72.0	0.001
50–59	66.7	82.6	48.3	65.9	<0.001
60–69	43.3	50.0	57.7	50.3	0.13
70–79	61.2	50.6	50.0	53.9	0.2
80–90	56.8	44.3	53.1	51.4	0.19
Cared for at the habitual hospital, %	99.0	99.3	96.0	98.4	0.011
Occupational status, %					-
Medical leave due to PD	1.48	3.70	5.29	3.12	
Medical leave due to other reason	0.74	1.35	0.00	0.75	
Employed	5.93	10.1	11.9	8.72	
Unemployed	0.74	1.01	0.88	0.86	
Retired	74.6	56.9	71.8	68.2	
Has never worked	12.8	19.9	10.1	14.4	
Other/unknown	3.07	7.07	0.00	3.88	
Disease duration					0.006
Mean ± SD, years	6.07 ± 5.42	7.51 ± 6.18	6.40 ± 5.47	6.61 ± 5.72	
<10 years, %	77.0	70.7	74.0	74.2	
≥10 years, %	23.0	29.3	26.0	25.7	
Hoehn and Yahr stage in ON state, n (%)					0.002
1	74 (19.7)	47 (16.9)	14 (6.80)	135 (15.7)	
2	153 (40.8)	112 (40.3)	103 (50.0)	368 (42.8)	
3	102 (27.2)	96 (34.5)	61 (29.6)	259 (30.1)	
4	40 (10.7)	20 (7.2)	25 (12.1)	85 (9.90)	
5	6 (1.60)	3 (1.08)	3 (1.46)	12 (1.40)	
Premotor symptom, ^b^ n (%)					
Hyposmia	110 (37.4)	72 (33.2)	54 (28.9)	236 (33.8)	0.015
REM sleep disorder	94 (29.6)	65 (28.9)	60 (30.9)	219 (29.7)	0.09
Depression	143 (42.9)	96 (39.2)	74 (38.3)	313 (40.6)	0.71
Constipation	155 (59.2)	103 (44.8)	102 (54.8)	360 (49.7)	0.14
Concomitant disease, %	46.9	34.3	35.7	40.1	0.001
Charlson comorbidity index ^c^					<0.0001
Mean ± SD	3.81 ± 1.74	3.01 ± 1.45	3.43 ± 1.71	3.46 ± 1.68	
Median	4	3	3	3	
Q1–Q3	3–5	2–4	2–4	2–4	
Min–Max	0–11	0–7	0–10	0–11	
10-year survival probability, %					<0.0001
Mean ± SD	54 ± 31	69 ± 27	62 ± 30	61 ± 30	
Median	53	77	77	77	
Q1–Q3	21–77	53–90	53–90	53–90	
Min–Max	0–98	0–98	0–98	0–98	

Max, maximum; Min, minimum; PD, Parkinson’s disease; Q, quarter; REM, rapid eye movement; SD, standard deviation. ^a^ Percentage of males per decade of age. ^b^ Number and percentage of patients who at the time of diagnosis reported having had the specific premotor symptom. This information was not available for 123 patients; thus, these patients were excluded. ^c^ Age-adjusted.

**Table 2 brainsci-11-01557-t002:** Clinical and sociodemographic characteristics of patients with non-APD and APD, according to hospital type.

	Hospital with General Neurology Service (n = 405)		Hospital with Parkinson’s Disease Dedicated Clinic (n = 297)		Hospital with Movement Disorder Unit (n = 227)		Total Non-APD(n = 574)		*p* Value ^a^
Characteristic	Non-APD (n = 251)	APD (n = 154)	Non-APD (n = 185)	APD (n = 112)	Non-APD (n= 138)	APD (n= 89)		Total APD (n = 355)	
Age, mean ± SD, years	72.2 ± 9.8	75.6 ± 10.0	68.0 ± 9.9	71.7 ± 9.2	71.4 ± 9.9	71.4 ± 9.9	70.7 ± 10.0	73.7 ± 10.0	<0.0001
Sex, female, %	43.8	53.2	41.1	48.2	44.9	50.6	43.2	51.0	0.021
Occupational status, %									-
Medical leave due to PD	0.80	2.60	3.78	3.57	3.62	7.87	2.44	4.23	
Medical leave due to other reason	0.80	0.65	1.62	0.89	0.00	0.00	0.75	0.87	
Employed	7.97	2.60	14.6	2.68	16.7	4.49	12.2	3.10	
Unemployed	1.20	0.00	0.00	2.68	0.72	1.12	0.70	1.13	
Retired	72.9	77.3	53.5	62.5	73.2	69.7	66.7	70.7	
Has never worked	12.75	13.0	17.8	23.2	5.80	16.8	12.7	17.2	
Other/unknown	3.59	3.90	8.64	4.47	0.00	0.00	4.35	3.10	
Disease duration, mean ± SD, years	3.92 ± 3.70	9.59 ± 5.95	4.69 ± 3.99	12.2 ± 6.35	3.37 ± 2.98	11.1 ± 5.1	4.04 ± 3.67	10.8 ± 6.0	<0.0001
Hoehn & Yahr stage in ON state ^b^, median	2	3	2	3	2	3	2	3	<0.0001
Premotor symptom at diagnosis ^c^, %									
Hyposmia	36.6	39.2	36.6	26.4	28.9	28.8	27.9	21.5	<0.0001
REM sleep disorder	27.3	33.9	24.2	38.9	24.0	43.5	21.6	26.8	<0.0001
Depression	37.1	53.3	36.0	45.7	30.7	52.2	30.7	38.7	<0.0001
Constipation	40.6	68.2	39.0	56.6	47.1	69.2	34.7	45.8	<0.0001
Concomitant disease, %	45.0	50.0	34.0	34.8	36.2	34.8	39.4	41.4	0.54
Charlson comorbidity index ^d^, mean ± SD	3.60 ± 1.68	4.15 ± 1.78	2.84 ± 1.39	3.29 ± 1.52	3.25 ± 1.52	3.71 ± 1.95	3.27 ± 1.58	3.77 ± 1.78	<0.0001
10-year survival probability, median, %	53	53	77	77	77	77	77	53	<0.0001
APD prevalence, relative frequency %	61.9	38.1	62.3	37.7	60.8	39.2	61.8	38.2	NA

APD, advanced Parkinson’s disease; PD, Parkinson’s disease; REM, rapid eye movement; SD, standard deviation. ^a^ Comparison between non-APD and APD patients. ^b^ At the time of inclusion in the study. ^c^ This information was not available for 123 patients; thus, these patients were excluded. ^d^ Age-adjusted. Only 54 out of 355 patients with APD (15.2%) were receiving some DAT. Statistically significant differences regarding the use of DATs in patients with APD were found between different hospital types (i.e., 21.3% of patients in hospitals with movement disorder units received DATs compared with 18.7% of patients in hospitals with PD dedicated clinics and 9.1% of patients in hospitals with general neurology services (*p* = 0.017)). Patients receiving DATs were treated for approximately 3 years (median treatment duration 3.0 years (Q1,Q3: 2.0,4.0 years)).

## Data Availability

AbbVie is committed to responsible data sharing regarding the clinical studies we sponsor. This includes access to anonymized, individual and trial-level data (analysis data sets), as well as other information (e.g., protocols and clinical study reports), as long as the studies are not part of an ongoing or planned regulatory submission. This includes requests for clinical studies data for unlicensed products and indications. This clinical study data can be requested by any qualified researchers who engage in rigorous, independent scientific research, and will be provided following review and approval of a research proposal and statistical analysis plan (SAP) and execution of a data sharing agreement (DSA). Data requests can be submitted at any time and the data will be accessible for 12 months, with possible extensions considered. For more information on the process, or to submit a request, visit the following link: https://www.abbvie.com/our-science/clinical-trials/clinical-trials-data-and-information-sharing/data-and-information-sharing-with-qualified-researchers.html.

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
