# Peer review of "Prevalence of Advanced Parkinson’s Disease in Patients Treated in the Hospitals of the Spanish National Healthcare System: The PARADISE Study"

_brainsci, 2021, doi:10.3390/brainsci11121557_

Round 1

Reviewer 1 Report

In the present large, cross-section, multicenter national study, Martinez-Castrillo and colleagues describe the prevalence and incidence of patients with a diagnosis of advanced Parkinson’s disease according to the neurologists criteria (gold standard) and the outcome of the Questionnaire for Advanced Parkinson’s Disease (CDEPA), which was released previously. The study included a total of 929 patient from three types of hospitals, whose distribution represented the total Spanish population, and found that about 38% of PD patients are at advanced stage, without differences among the three types of hospital. The authors describe the frequency of patients treated with device-aided therapies (deep brain stimulation, levodopa/carbidopa intestinal gel, continuous apomorphine subcutaneous infusion)

Major:

  • The definition of advanced PD should be provided in the abstract
  • The authors should clarify what are the differences between a ‘movement disorders unit’ and a ‘PD dedicated’ unit.
  • The relative frequency of APD patients should be provided in table 2.
  • The analysis of sex distribution should be adjusted by age.
  • It is unclear why the authors used two definitions ‘device-aided therapies’ and ‘second-line therapy’. Is there any difference? If so, specify. I suggest to consistently use one acronym thoughout the manuscript, figures and legends and I discourage the use of the term ‘DAT’ because this is commonly used for dopamine transporter.
  • Figure 1 should be edited: (i) there are 6 sections in the figure but the legend includes only 4 options; (ii) some colors cannot be clearly differentiated.
  • It would be interesting to know the different approach to the three types of device-aided therapies (DBS, LCIG, CSAI) among the three different types of hospitals.
  • It is pretty obvious and pleonastic that PDQ-8 is significantly worse among APD nAPD. It may be more interesting to know which items of the PDQ-8 are more affected by this dichotomy and wich remain similar between APD and nAPD.
  • Diagnostic criteria for advanced PD may be more clear if authors include the CDEPA among supplementary documents.

Minor:

  • The manuscript should be thoroughly revised by an native English speaking. This revision should include (i) editing ‘an DAT’ into ‘a DAT’ throughout the manuscript, (ii) rephrasing the repetition of the same sentence ‘clinical and sociodemographic differences between patients …’ in the first and second paragraphs of the discussion.

Reviewer 2 Report

This study presents the prevalance of patients with advanced Parkinson's Disease (APD). Authors presents the results based on the examinations performed in Spain in various types of departments. The study aknowledges an important topic, as the examination, treatment and care concerning APD patients is an evolving issue, however there are certain points, which in my opinion should be addressed by the authors:

  1. Authors state "In this advanced stage, the disease is highly disabling, has a complex management, impacts seriously patient’s quality of life (QoL), and increases the Healthcare System expenses", which is true. In order to provide a sufficient introductory statement I believe that a brief statement concerning the disabling features in PD would be beneficial. Moreover in the context of quality of life and Healthcare System expenses, I think that the underscored issue is  the correlation of PD and cancer incidence, which could be also mentioned (Ejma et al, 2020). Adequately other features not directly related to PD symptomatology, however associated with higher incidence in PD could be stated.
  2.  "Sociodemographic data and clinical variables [age, sex, occupational status, PD disease duration, Hoehn and Yahr (H&Y) stage at the time of PD diagnosis and at the time of patient inclusion in the study, pre-motor symptoms, and concurrent comorbidities]
    were obtained from computerized hospital records, patient medical records, or discharge reports" - the presence of which premotor symptoms was verified - only Hyposmia, REM sleep disorder
    Depression , Constipation
    as in the table?, was any cognitive assesment performed? what comorbidities were detected? was the treatment of patients a factor affecting inclusion to the study?
  3. Was any neuroimaging performed? Due to overlapping manifestation in APD, I believe that possible errors in the examination concerning possible PSP-P should be mentioned in the limitation section (Frontiers in Neurology 2020, doi: 10.3389/fneur.2020.00180)
  4. "Nevertheless, we consider that this patient population exclusion had little impact on the results obtained as only 3.8% of the Spanish population were cared in private centers when the study begun." (Reference should be added)
  5. A separate section concerning limitations could facilitate interpretation. Additionally I suggest implementing a table with the listed limitations and the comments of the authors.

Round 2

Reviewer 1 Report

The authors have satisfactorily replied to all the issues raised. I have no further comment.

Reviewer 2 Report

The authors have satisfactorily addressed my comments.